# Cdh1 Deficiency Sensitizes TNBC Cells to PARP Inhibitors

**DOI:** 10.3390/genes13050803

**Published:** 2022-04-30

**Authors:** Junjun Li, Mengjiao Lan, Jin Peng, Qunli Xiong, Yongfeng Xu, Yang Yang, Ying Zhou, Jinlu Liu, Zhu Zeng, Xiaojuan Yang, Zhiwei Zhang, Pumin Zhang, Qing Zhu, Wei Wu

**Affiliations:** 1Department of Abdominal Oncology, West China Hospital, Sichuan University, Chengdu 610000, China; lijunjun9862@126.com (J.L.); xqunly@163.com (Q.X.); hxxyf2020@163.com (Y.X.); yangyangmm2022@163.com (Y.Y.); zhou911550@outlook.com (Y.Z.); liujinlu@stu.scu.edu.cn (J.L.); zengzhuwakeup@163.com (Z.Z.); y3532388972@163.com (X.Y.); hxlczhangzhiwei@163.com (Z.Z.); 2Zhejiang Provincial Key Laboratory of Pancreatic Disease, The First Affiliated Hospital of Zhejiang University, Hangzhou 310003, China; lanmengjiao1042@126.com (M.L.); pengjin0000@126.com (J.P.); pzhangbcm@zju.edu.cn (P.Z.)

**Keywords:** Cdh1, BRCA1, triple-negative breast cancer, PARP, olaparib

## Abstract

Triple-negative breast cancer (TNBC) is a type of breast tumor that currently lacks options for targeted therapy. Tremendous effort has been made to identify treatment targets for TNBC. Here, we report that the expression level of anaphase promoting complex (APC) coactivator Cdh1 in TNBC is elevated compared to that in the adjacent healthy tissues, and high levels of Cdh1 expression are correlated with poor prognoses, suggesting that Cdh1 contributes to the progression of TNBC. Interfering with the function of Cdh1 can potentiate the cytotoxic effects of PARP inhibitors against BRCA-deficient and BRCA-proficient TNBC cells through inducing DNA damage, checkpoint activation, cell cycle arrest, and apoptosis. Further investigation reveals that Cdh1 promotes BRCA1 foci formation and prevents untangled DNA entering mitosis in response to PARP inhibition (PARPi) in TNBC cells. Collectively, these results suggest that APC/Cdh1 is a potential molecular target for PARPi-based therapies against TNBCs.

## 1. Introduction

Breast cancer is the most diagnosed cancer and the leading cause of cancer-related death in women, with an estimated 2.3 million new cases (11.7%) and 684,996 deaths (6.9%) reported in 2020 [1]. Metastasis, recurrence, and drug resistance are common causes of breast cancer treatment failure. Breast cancer can be classified into four subtypes: HER2-enriched, luminal A, luminal B, and triple-negative breast cancer (TNBC) [2]. Treatment options include mastectomy, chemotherapy (CT), hormone therapy (HT), and radiotherapy (RT) [3]. However, because of high mortality rates for breast cancer, there is an ongoing effort to improve current therapies and find new therapies.

TNBC accounts for 10–20% of all breast cancer cases [4]. It is a heterogeneous disease that lacks the expression of human epidermal growth factor receptor 2 (HER2), estrogen receptor (ER), and progesterone receptor (PR), and currently, there are no targeted therapies available. TNBCs respond to chemotherapies initially but quickly develop resistance [5]. Therefore, TNBC is more aggressive and has a poorer prognosis compared to other breast cancers.

Poly-ADP-ribose polymerases (PARPs) are enzymes important for the repair of single-stranded (SSBs) and double-stranded (DSBs) DNA breaks, and for the stabilization of DNA replication forks [6]. Certain cancer cells, including TNBC cells with *BRCA1/2* mutations, are hypersensitive to PARP inhibition, although the underlying mechanisms are still poorly understood [7]. However, *BRCA1/2*-mutated TNBCs constitute only approximately 20% of all TNBC cases [8]. Therefore, there is an urgent need to identify ways to realize the benefits of PARPi treatment in non-BRCA mutant TNBCs.

The anaphase-promoting complex/cyclosome (APC/C) is an E3 ubiquitin ligase that targets substrates for proteasomal degradation [9]. APC/C utilizes two adaptor proteins, Cdc20 and Cdh1 (Fzr1); Cdc20 activates APC/C in early mitosis until anaphase, and Cdh1 acts in the G1 phase [10]. Furthermore, the activation of APC/C has been implicated in response to X-irradiation-induced DNA damage [11]. Our previous work showed that Cdh1 is required for the cells to choose homologous recombination (HR)-mediated DSB repair over non-homologous end joining (NHEJ) due to its function in mediating the ubiquitination and degradation of USP1, a deubiquitinase that removes ubiquitin chains on DNA damage signaling molecules [12]. In addition, APC/Cdh1 is required to safeguard genome stability in response to replication stress (RS) [13,14]. Therefore, tumor cells lacking Cdh1 are very likely sensitive to DNA damage and to drugs that induce replication stress such as PARPi.

RS is manifested as replication fork slowing or stalling and is a byproduct of tumor development. On one hand, it is a known driver of genome instability that can promote tumor progression [15]. On the other hand, excessive RS can cause massive DNA breakage and cell death, impeding tumor growth [16,17]. Therefore, targeting RS is one of the most used strategies in cancer therapy [18]. Recent findings indicate that TNBC exhibits a high level of RS [19,20]. Targeting the proteins involved in RS response alone or in combination with RS inducers, such as PARP inhibitors, might specifically kill TNBC cells.

In this study, we show that Cdh1 is highly expressed in tumor tissues and negatively correlated with the overall survival in the TNBC group. The depletion of Cdh1 expression leads to an increased sensitivity to PARP inhibition and DNA damage in TNBC cells. Our results show that Cdh1 depletion may be a promising strategy to improve the efficacy of PARP inhibitors in TNBC treatment.

## 2. Materials and Methods

### 2.1. Cell Culture

MDA-MB-231, MDA-MB-468, HCC 1806, and T47D cell lines were purchased from the American Type Culture Collection (ATCC; Rockefeller, Maryland, USA). The cells were cultured in either DMEM, DMEM/F12 (50:50), or RPMI-1640 media supplemented with 10% FBS, 1 μg/mL penicillin, and streptomycin. The medium, fetal bovine serum (FBS), trypsin, and penicillin-streptomycin were purchased from Gibco (Grand Island, NY, USA). All cells were maintained in an incubator supplemented with 5% CO_2_ at 37 °C.

### 2.2. Western Blot Analysis

After the indicated treatment, the cells were washed twice with cold PBS buffer and lysed with cold RIPA lysis buffer (Applygen Technologies Inc., Beijing, China) supplemented with protease and phosphatase inhibitors (Roche Diagnostics, Mannheim, Germany). Equal amounts of total proteins were resolved on a 10% SDS polyacrylamide gel and transferred onto a polyvinylidene difluoride membrane (PVDF, Billerica, MA, USA). The membranes were incubated for 1 h in blocking buffer (5% nonfat dry milk in TBST) and subsequently incubated with primary antibodies overnight at 4 °C. The membranes were washed with TBST and incubated for 1 h with an anti-rabbit or anti-mouse antibody at 25 °C. After washing with TBST three times, the membranes were visualized and analyzed using the chemiluminescent imaging system LAS 500 (GE Healthcare, Boston, MA, USA) according to the standard procedures. The expression levels of actin or GAPDH were routinely used as a loading control.

### 2.3. Assays for Cell Proliferation

For the MTS assay, cells were trypsinized using an EDTA/trypsin solution, seeded in 96-well plates at a density of 3000 cells per well, and then cultured for the indicated time periods. At the end of the incubation period, the number of viable cells was determined using a colorimetric assay (MTS; Promega, Madison, WI, USA). In brief, the culture medium was removed, and 100 μL fresh complete culture medium was added to 20 μL of MTS in each well. The cells were then incubated for 2 h before the absorbance of the formazan product was measured (λ = 490 nm).

### 2.4. Plasmids and Lentiviruses

Plasmids used in this study were generated using standard cloning methods. ShRNAs were constructed in Tet-on pLKO.1 with the following sequences (Table 1).

The shRNA-carrying lentiviruses used for obtaining Cdh1 knockdown and negative control cells were produced in the laboratory. The infected cells were selected by the puromycin treatment (4 μg/mL) method for 2 days. Next, we generated a stable doxycycline-inducible lentivirus shRNA vector to knock down Cdh1 in breast cancer cell lines.

### 2.5. Antibodies

The antibodies used in this study were as follows: anti-Cdh1 (sc-56312, 1:1000 WB, Santa Cruz Biotechnology, Santa Cruz, CA, USA); anti-γH2AX (05-636, 1:500 IF, 1:1000 IF, Millipore, Billerica, MA, USA); anti-BRCA1 (sc-6954, 1:200 IF, Santa Cruz); anti-Phospho-Chk1(Ser345) (2348S, 1:1000 WB, Cell Signaling, Danvers, MA, USA); anti-cleaved Caspase-3 (9664S, 1:1000 WB, Cell Signaling); anti-cleaved PARP (9664S, 1:1000 WB, Cell Signaling); anti-Actin (66009-1-Ig, 1:5000 WB, Proteintech, Wuhan, China); anti-GAPDH (60004-1-1g, 1:5000 WB, Proteintech, Wuhan, China), and anti-PICH(8886S, 1:200 IF, Cell Signaling); the secondary antibodies conjugated to horseradish peroxidase were used for Western blotting. The secondary antibodies of anti-mouse, or anti-rabbit containg Alexa Fluor 488 or 594 were used for immunofluorescence staining (Jackson ImmunoResearch Laboratories, West Grove, PA, USA).

### 2.6. Immunostaining

After the indicated treatment, cells were plated on coverslips, fixed with 4% paraformaldehyde for 15 min, washed with cold PBS buffer twice, permeabilized in 0.5% Triton X-100 for 5 min, blocked with 5% BSA in PBS buffer for 1 h at room temperature, and then incubated with primary antibodies at 4 °C overnight. After washing them three times in cold PBS, they were incubated with secondary antibodies for 30 min at 37 °C. All images were obtained using a Nikon Ni-E microscope (Nikon Corporation, Tokyo, Japan) with identical exposure times for each sample.

### 2.7. Ultrafine Anaphase Bridges Detection

The detection of Plk1-interacting checkpoint helicase (PICH)-coated ultrafine anaphase bridges (UFBs) was performed according to a previously published protocol [21].

### 2.8. Apoptosis Assay

Apoptotic cells were determined using the Annexin V-FITC/PI Apoptosis Kit (MultiSciences, China) according to the manufacturer’s instructions. Briefly, cells were trypsinized, washed, and resuspended in 500 μL of binding buffer. Then, 10 μL FITC Annexin V and 5 μL PI were added to the cell suspension. The cells were incubated in the dark for 5 min and analyzed using BD FACSCanto II.

### 2.9. Cell Cycle Analysis

Cell cycle analysis was performed using the Cell Cycle Staining Kit (MultiSciences, China). Breast cancer cells were washed with cold PBS buffer three times, fixed in 70% ethanol at −20 °C for 12 h, washed with cold PBS buffer, and stained with 0.5 mL of a propidium iodide (PI) staining buffer containing 200 mg·mL^−1^ RNase A and 50 μg·mL^−1^ PI, at 37 °C for 30 min in the dark. Analyses were performed using BD FACS Canto II.

### 2.10. Clinical Samples and Data Acquisition

TCGA RNA-seq datasets for TNBC were obtained from the Gene Expression Omnibus (GEO) database (http://www.ncbi.nlm.nih.gov/geo (accessed on 30 September 2021)). GSE115275 [22] was downloaded from the GEO database. The GSE115275 datasets contained six TNBC tissues and six adjacent normal, healthy tissues.

### 2.11. Analysis of Differentially Expressed Genes between TNBC and Normal Tissue Samples

Differential expression analysis (DEG) was performed using the Linear Models for Microarray Data (LIMMA) package. Samples were separated into TNBC and normal, healthy tissue groups. The adjusted *p*-value was analyzed to correct for false positive results in the GEO datasets. Having an adjusted *p*-value < 0.05 and a |log2FC| > 1 were defined as the thresholds for screening differential expression of mRNAs. The Limma package (version: 3.40.2) of R software was used to analyze the differential expression of mRNAs. The R software ggord package was used to draw the PCA plot, and pheatmap the package was used to draw a heatmap.

### 2.12. Statistical Analysis

The results are presented as the mean ± SD. The data were analyzed using GraphPad Prism 9.0 and ImageJ. In all cases, differences were considered statistically significant if the *p* value was < 0.05 (*, *p* < 0.05; **, *p* < 0.01; ***, *p* < 0.001, ****, *p* < 0.0001).

## 3. Results

### 3.1. Cdh1 Expression Correlates with Poor Prognosis in TNBC

To determine if the expression of Cdh1 is correlated with breast cancer, we analyzed the GSE115275 dataset [22]. Through principal component analysis (PCA), it is clear that the normal and TNBC tissues could be well separated into two groups without any intersection (Figure 1A). With a cut-off of two-fold change of expression plus *p*-value < 0.05, 5644 DEGs could be identified in the breast tumor tissues compared to the normal tissues (Figure 1B). A heat map was used to reveal the top genes ranked by *p*-value and fold change (FC). The results indicated that the expression profiles differed significantly between cancerous and healthy breast tissues. We found that Cdh1 (FZR1, fizzy-related 1) was highly expressed in the tumor tissues (Figure 1C). Given the role of Cdh1 in DNA damage repair [23,24], we next sought to determine if its expression is correlated with survival after chemotherapy. Using the publicly available database (https://kmplot.com/analysis (accessed on 14 April 2022)), we found that the expression of Fzr1 (Cdh1) was negatively correlated with the overall survival rate in TNBC patients (Figure 1D,E).

### 3.2. Cdh1-Deficient TNBC Cells Are Sensitive to the PARP Inhibitor Olaparib

Given that APC/Cdh1 is critical to the function of HR repair factors [12], we hypothesize that tumor cells lacking Cdh1 will be as sensitive to replication pressure as cells without BRCA1. Thus, we hypothesize that Cdh1 deletion can greatly improve the sensitivity of tumor cells to PARP inhibitors. To that end, we first examined whether Cdh1 played any role in the proliferation of TNBC cells. We used two different shRNAs to deplete Cdh1 expression in a BRCA-mutant TNBC cell line (HCC 1806), two BRCA-wide type cell lines (MDA-MB-231 and MDA-MB-468), and a non-TNBC cell line (T47D). Western blot analysis showed that Cdh1 could be largely depleted from these cell lines (Figure 2A). However, Cdh1 depletion did not interfere with the proliferation of breast cancer cells (Figure 2B). We then investigated the effects of Cdh1 knockdown in conjunction with the use of a PARP inhibitor (olaparib). As shown in Figure 2C, a 96 h olaparib treatment resulted in a dose-dependent suppression of proliferation in these cell lines. As expected, Cdh1-depleted cells became more sensitive to olaparib than the control cells. Notably, this effect is much more pronounced in the TNBC cells than in the non-TNBC cells (T47D) (Figure 2C).

### 3.3. PARP Inhibitor Induces Elevated Levels of Apoptosis and G2-M Cell Cycle Arrest in Cdh1 Depleted TNBC Cells

Next, we sought to determine the effects of olaparib on the Cdh1-depleted cells. The results from the apoptosis assay showed that a 96 h PARPi treatment could induce apoptosis in all tested cell lines. Importantly, Cdh1 depletion can substantially increase the apoptotic effect of PARPi on TNBC cells, but not on non-TNBC (T47D) cells (Figure 3A,B). Furthermore, cell cycle analysis of MDA-MB-468 cells suggested that PARPi significantly increased the proportion of cells in the G2-M phase of the cell cycle in Cdh1 depleted cells (Figure 3C,D). Consistently, caspase-3 activation and proteolytic cleavage of PARP-1 were apparent in PARPi treated Cdh1 deficient cells (Figure 3E). These results indicated that the anti-proliferation effect of PARPi in Cdh1 depleted TNBC cells was associated with apoptosis and G2-M phase cell cycle arrest.

### 3.4. PARPi Causes DNA Damage in Cdh1 Depleted TNBC Cells

G2-M phase cell cycle arrest and apoptosis are normally associated with increased DNA damage. To assess whether PARPi can induce DNA damage in Cdh1 depleted TNBC cells, we performed immunofluorescence by using two DSB surrogate markers: γH2AX and 53BP1. The results showed that the number of PARPi induced γH2AX foci in Cdh1 depleted cells was increased more than that in control cells in MDA-MB 468 and MDA-MB 231 cells (Figure 4A,B). Similarly, the number of PARPi-induced 53BP1 foci in Cdh1 depleted cells increased more than that in control cells in MDA-MB-468 and MDA-MB-231 cells, respectively (Figure 4A,B). Western blotting analysis in MDA-MB-468 cells also indicated that PARPi could induce substantially increased levels of γH2AX and cause checkpoint activation in Cdh1 depleted cells (Figure 4C,D). In conclusion, PARPi can elicit increased levels of DNA damage and checkpoint activation in Cdh1-depleted TNBC cells.

### 3.5. Cdh1 Promotes BRCA1 Foci Formation and Prevents Untangled DNA Entering Mitosis in Response to PARPi in TNBC Cells

We next investigated the possible mechanisms by which Cdh1 augments the sensitivity to PARPi in TNBC cells. We previously showed that Cdh1 is required for efficient HR through facilitating BRCA1 recruitment to DSBs that are caused by microirradiation or prolonged hydroxyurea (HU) treatment [12]. Our immunofluorescence analysis showed that PARPi could substantially increase BRCA1 foci formation, indicating that PARPi-induced DNA damage might need BRCA1 to repair. Most importantly, when Cdh1 was depleted, PARPi-induced BRCA1 foci was substantially compromised, suggesting that Cdh1 is required for BRCA1’s foci formation in response to PARPi (Figure 5A,B).

Our cell proliferation assay (Figure 2C) showed that both *BRCA1/2* wild-type (MDA-MB-231) and *BRCA1/2* mutant (HCC 1806) TNBC cell lines were sensitive to PARPi when Cdh1 was depleted, suggesting that Cdh1 depletion can cause PARPi sensitivity independent of its effect on BRCA1. Recent findings suggested that mitotic progression is responsible for PARPi cytotoxicity [25]. Mechanically, when BRCA1/2 is absent, PARPi induces entangled DNA (most likely originating from aberrant replication intermediates) that are transmitted into mitosis, which would affect anaphase chromosome separation leading to genome instability and cell death in G1 daughter cells. To determine whether Cdh1 depletion can induce entangled DNA persisting into mitosis upon PARPi treatment, we performed immunofluorescence with MDA-MB-231 and HCC1806 cells to check PICH (Plk1-interacting checkpoint helicase)-coated ultrafine DNA bridges (UFBs), which are known to be caused by entangled DNA [26]. The results showed that, in both cell lines, olaparib treatment could induce a significant increase in PICH-coated UFBs when Cdh1 was absent (Figure 5C,D), indicating that Cdh1 plays an important role in preventing the pathological accumulation of untangled DNA entering mitosis to preserve genomic stability in TNBC cells.

## 4. Discussion

TNBC represents approximately 30% of breast cancer-associated deaths. With no positive markers for classification and a lack of specific treatment targets such as ER or HER2, there remains a major problem in the treatment of TNBC [27]. PARP inhibitors are expected to provide a novel therapeutic strategy in the treatment of TNBC, and the use of such compounds has shown a significant clinical benefit in patients with BRCA-deficient TNBC [28,29]. However, carriers of BRCA mutations account for only a fraction of patients with TNBC, and attention still needs to be paid to the treatment of BRCA-proficient TNBC [30].

In this study, we demonstrated that Cdh1 is highly expressed in tumor tissues and negatively correlated with overall survival in the TNBC groups (Figure 1). Although Cdh1 depletion had a negligible effect on the proliferation of breast cancer cell lines used in this study (Figure 2B), it can significantly potentiate the cytotoxic effects of PARP inhibitors on TNBC cells (Figure 2C). Flow cytometry and immunofluorescence analysis suggested that increased DNA damage, G2-M cell cycle arrest, and cell death were associated with the cytotoxic effects of PARPi in TNBC Cdh1 deficient cells (Figure 3 and Figure 4).

PARP inhibitors are particularly effective in killing BRCA mutation-associated cancer [7]. The synergistic effect of PARPi and Cdh1 depletion on DNA damage induction and cell proliferation inhibition suggests that Cdh1 and BRCA1/2 might play a similar role in repairing PARPi-induced DNA damage. Indeed, our published data showed that Cdh1 acts upstream of BRCA1 to promote efficient HR in repairing microirradiation or HU-induced DSBs [12]. In this study, we consistently observed that Cdh1 is also critical for PARPi-induced BRCA1 foci formation (Figure 5A), suggesting that Cdh1 and BRCA1 might function in the same pathway to repair PARPi induced DNA damage. Of note, in the BRCA1-mutated cell line HCC1806, Cdh1 depletion can further increase its sensitivity to olaparib, indicating that apart from its function in recruiting BRCA1 to damage sites, Cdh1 has other functions that are critical for cell survival following PARPi. Indeed, in response to replication stress, Cdh1 can prevent new origin firing or promote DNA lesion bypass to preserve genome stability [13,14]. In this study, we found that when Cdh1 was depleted, PARPi can induce a significant increase in entangled DNA entering into mitosis (Figure 5C), which is a known factor contributing to PARPi cytotoxicity [25]. Based on our data, we propose that Cdh1 is required for efficient recruitment of BRCA1 to PARPi-induced stalled or collapsed forks to prevent DSB formation. In addition, it has BRCA1-independent functions that are critical for fork stability. When Cdh1 is absent, PARPi induced stalled fork or collapsed fork could not be repaired properly, leading to DSB formation and cell death either in the first cell cycle or in the next cell cycle upon mitotic entry (Figure 6). In the future, it will be interesting to identify the source of entangled DNA generated upon Cdh1 depletion and PARPi, which will help to uncover the new functions of Cdh1 in DNA damage response and DNA repair. Overall, we demonstrate that Cdh1 is a potential carcinogenic promoter in TNBC progression. The overexpression of Cdh1 is associated with poor outcomes in patients with TNBC. Cdh1-depleted tumor cells, especially TNBC cells, are more sensitive to PARP inhibitors, highlighting the therapeutic potential of combining Cdh1 inhibition and PARPi for TNBC treatment.

## Figures and Tables

**Figure 1 genes-13-00803-f001:**
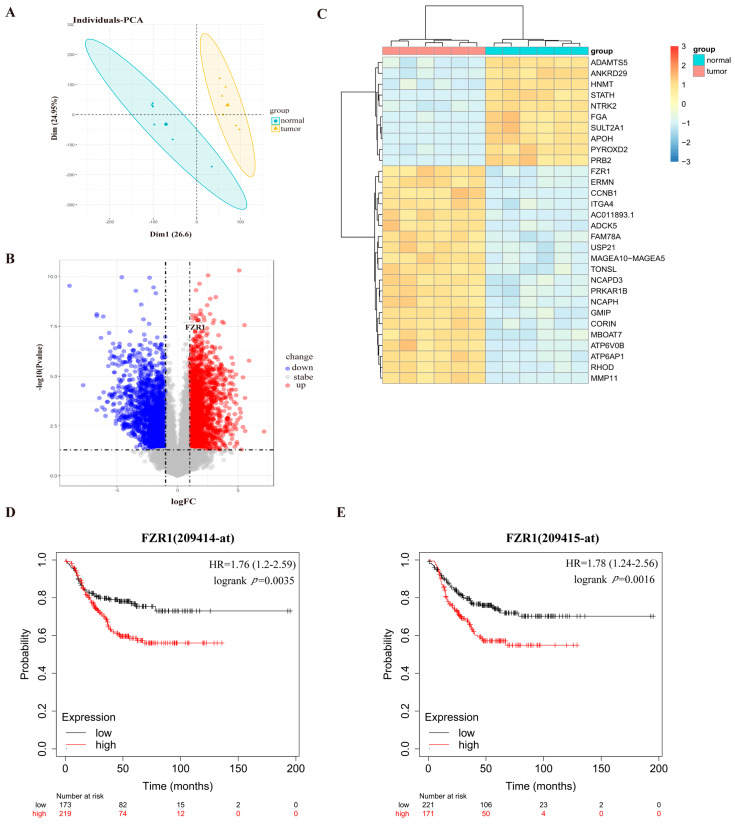
Cdh1 expression is elevated in breast cancer samples and correlated with poor survival in breast cancer patients. (**A**) PCA results before batch removal for multiple data sets. The normal and tumor data sets are separated without any intersection; (**B**) volcano plots were constructed using fold-change values and adjusted P. Blue/red plots indicate the down-regulated/upregulated DEGs with the cut-off criteria: |log2FC| > 1 and *p*-value < 0.05; (**C**) heatmap showing the differentially expressed mRNAs between tumor and normal tissues sorted by *p*-value (20 up-regulated genes and 10 down-regulated genes); (**D**,**E**) the KMPLOT analysis showed that Cdh1 negatively correlated prognosis in breast cancer patients. Red/Black lines indicate high/low expression of the corresponding genes.

**Figure 2 genes-13-00803-f002:**
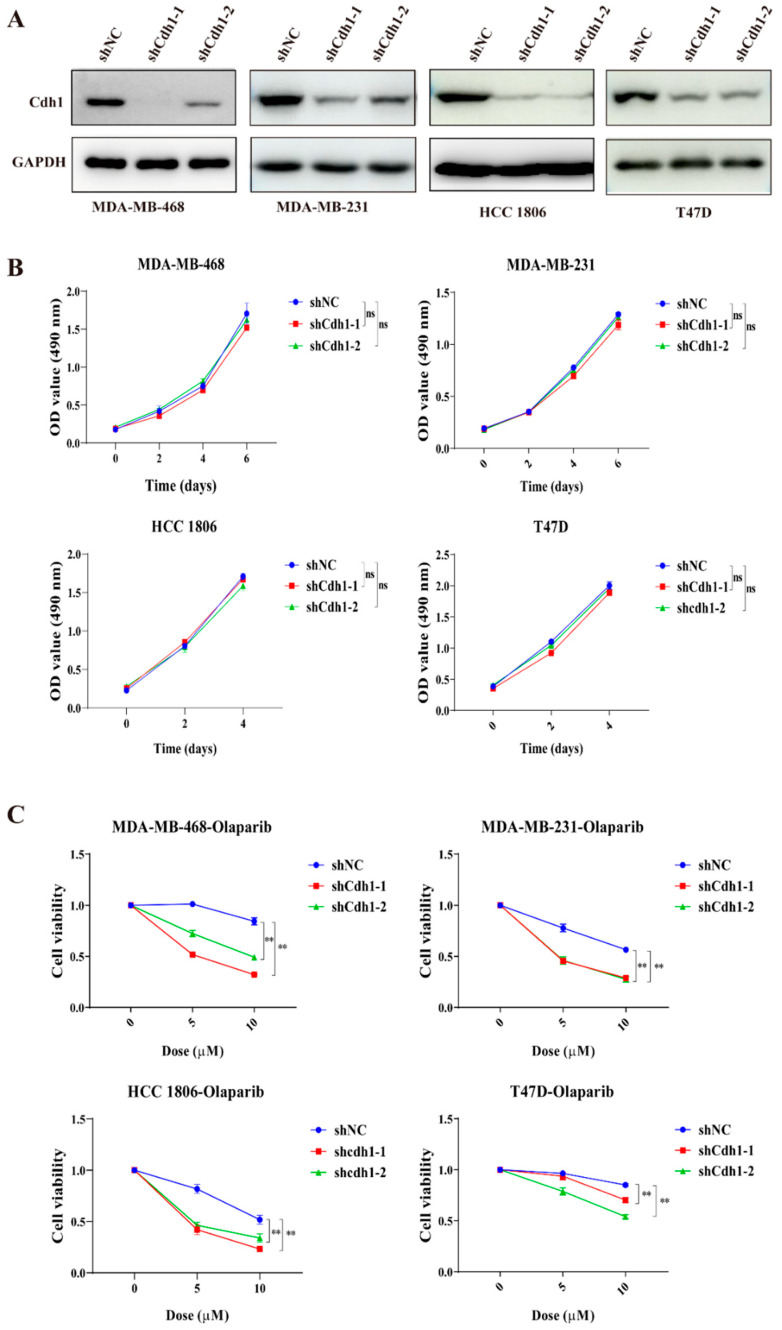
Cdh1 depletion sensitizes breast cancer cells to PARPi. (**A**) Western blotting analysis of Cdh1 in MDA-MB-468, MDA-MB-231, HCC 1806 and T47D cells with Cdh1 depletion via two independent shRNAs; (**B**) growth analysis of MDA-MB-468, MDA-MB-231, HCC 1806 and T47D cells with or without Cdh1 depletion; (**C**) Cell viability assay of MDA-MB-468, MDA-MB-231, HCC 1806 and T47D cells transduced with scrambled shRNA (NC) or shRNA against Cdh1 in response to Olaparib (0 μM 0, 5 μM and 10 μM) for 96 h, (** *p* < 0.01; ns: *p* ≥0.05).

**Figure 3 genes-13-00803-f003:**
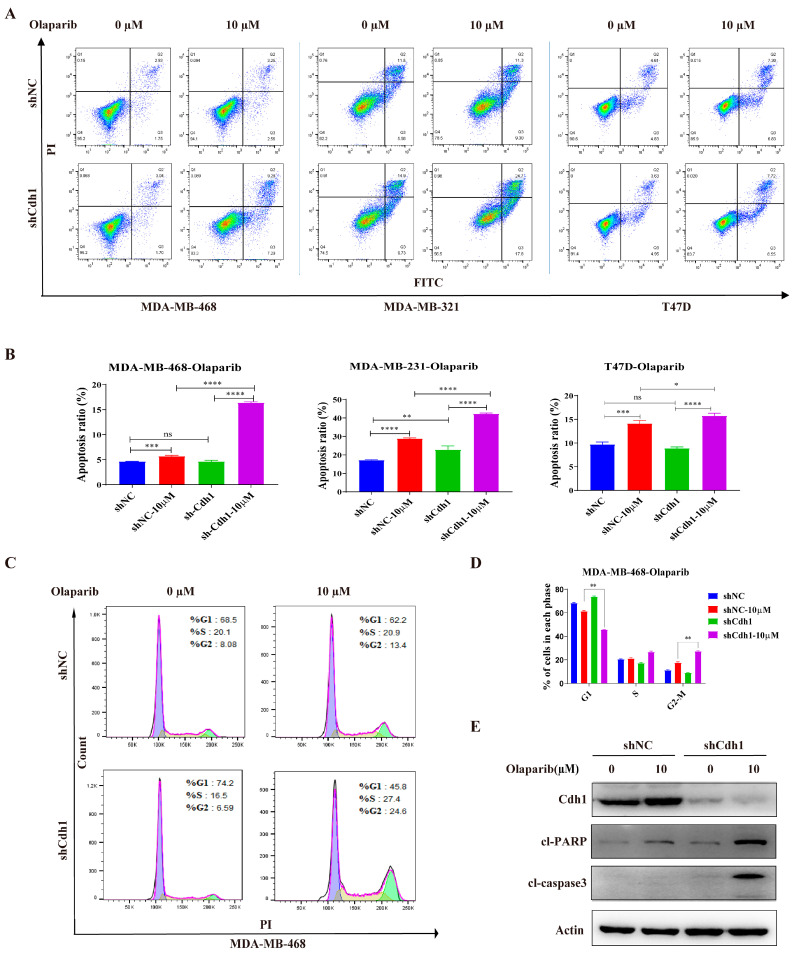
Cdh1 depletion induces apoptosis and G2-M arrest in TNBC cells following PARPi. (**A**) Flow cytometry analysis showing apoptosis of MDA-MB-468, MDA-MB-231 and T47D cells transduced with scrambled shRNA (NC) or shRNA against Cdh1 in in response to Olaparib (0 μM and 10 μM) for 96 h; (**B**) the percentage of apoptotic cells were quantified. Data are means of at least three independent experiments. Apoptosis rate was the sum of the two quadrats on the right in each FACS figure. Error bars indicate S.E.M. Significance of differences was calculated using Student’s *t* test (* *p* < 0.05, ** *p* < 0.01, *** *p* < 0.001, **** *p* < 0.0001; ns: *p* ≥ 0.05); (**C**) flow cytometry analysis showing cell cycle distribution of MDA-MB-468 cells transduced with scrambled shRNA (NC) or shRNA against Cdh1 in response to Olaparib (0 μM and 10 μM) for 96 h; (**D**) quantification of cell cycle distribution. Data are means of at least three independent experiments. Error bars indicate S.E.M. Significance of differences was calculated using Student’s *t* test (** *p* < 0.01; ns: *p* ≥ 0.05); (**E**) Western blotting analysis the expression of Cdh1, cl-PARP and cl-Caspase 3 in MDA-MB-468 cells transduced with scrambled shRNA (NC) or shRNA against Cdh1 in response to olaparib (0 μM and 10 μM) for 96 h.

**Figure 4 genes-13-00803-f004:**
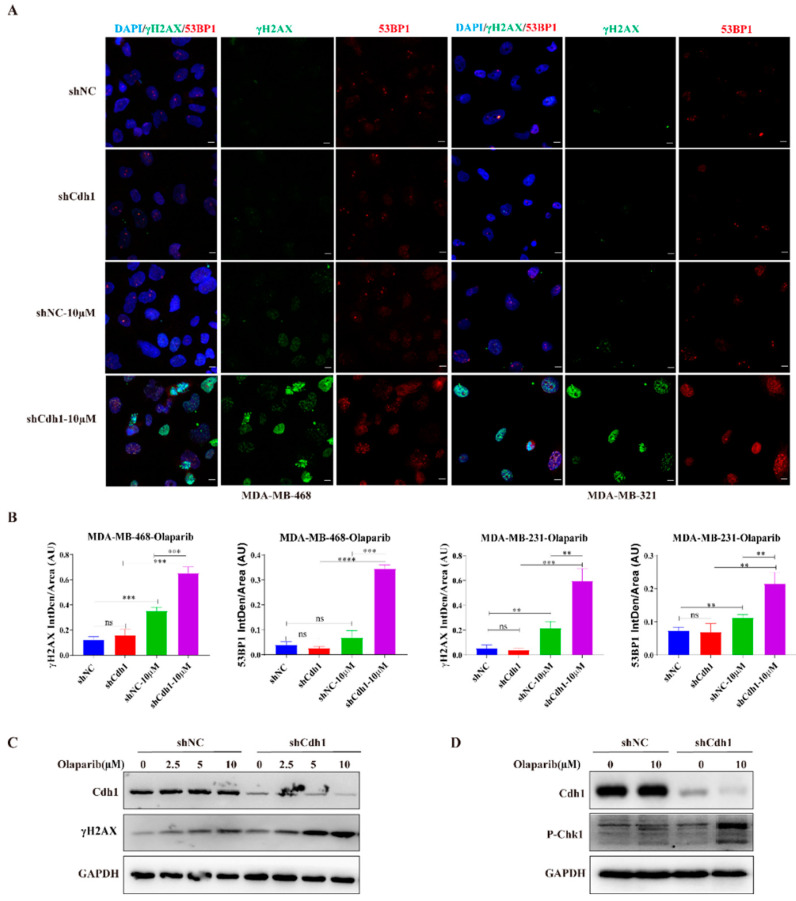
Cdh1 depletion induces DNA damage in TNBC cells following PARPi. (**A**) Immunofluorescence staining of γH2AX and 53BP1 in MDA-MB-468 and MDA-MB-231cells transduced with scrambled shRNA (NC) or shRNA against Cdh1 in in response to olaparib (0 μM and 10 μM) for 96 h, Scale bar: 50 μm; (**B**) the percentage of foci cells were quantified. Data are means of at least three independent experiments. Error bars indicate S.E.M. Significance of differences was calculated using Student’s *t* test (** *p* < 0.01, *** *p* < 0.001, **** *p* < 0.001; ns: *p* ≥ 0.05); (**C**) Western blotting analysis of the expression of Cdh1 and γH2AX in MDA-MB-468 cells transduced with scrambled shRNA (NC) or shRNA against Cdh1 in in response to olaparib (0 μM, 2.5 μM, 5 μM and 10 μM) for 96 h; (**D**) Western blotting analysis of the expression of Cdh1 and p-Chk1 in MDA-MB-468 cells transduced with scrambled shRNA (NC) or shRNA against Cdh1 in in response to olaparib (0 μM and 10 μM) for 96 h.

**Figure 5 genes-13-00803-f005:**
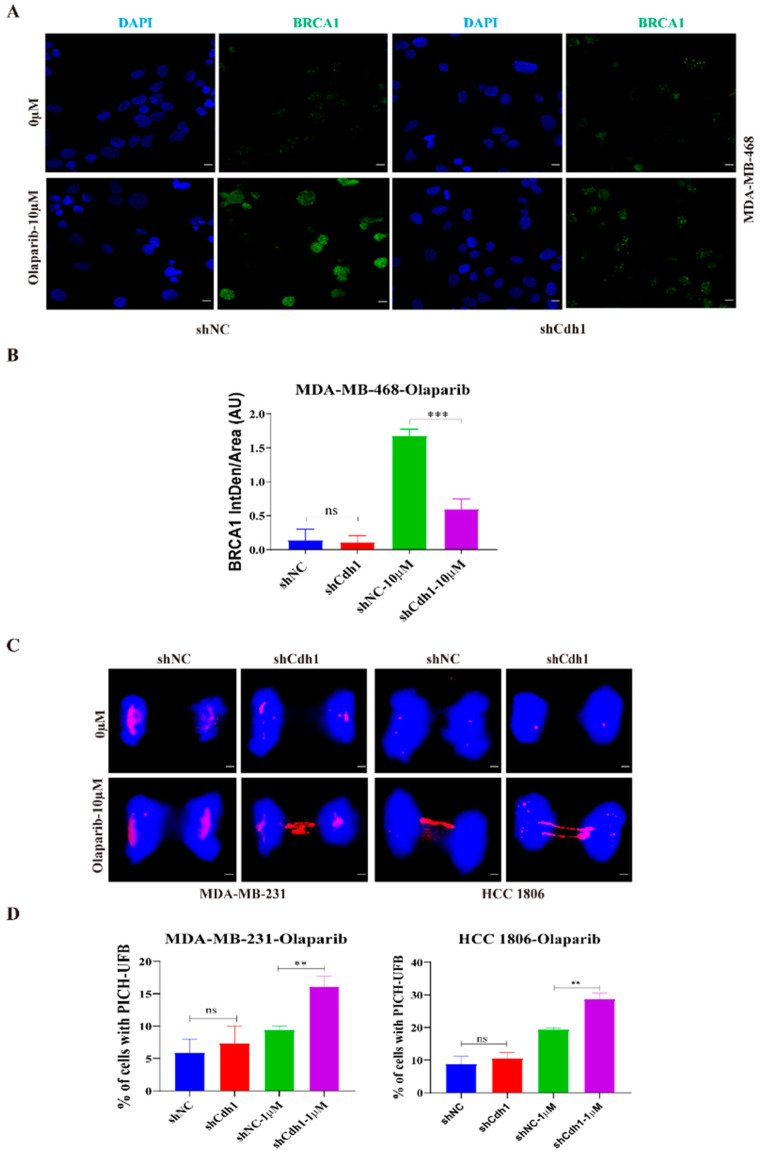
Cdh1 facilitates BRCA1 foci formation and prevents untangled DNA accumulation in mitosis following PARPi. (**A**) Immunofluorescence staining of BRCA1 in MDA-MB-468 cells transduced with scrambled shRNA (NC) or shRNA against Cdh1 in in response to olaparib (0 μM and 10 μM) for 96 h. Scale bar: 50 μm; (**B**) the percentage of BRCA1 foci cells were quantified. Data are means of at least three independent experiments. Error bars indicate S.E.M. Significance of differences was calculated using Student’s *t* test (*** *p* < 0.001; ns: *p* ≥ 0.05); (**C**,**D**) representative images (**C**) and quantification (**D**) of UFB at anaphase cells. PICH and DNA were visualized using anti-PICH antibody (red), and DAPI (blue), respectively. Scale bars, 10 μm. Data are means of three independent experiments. Error bars indicate S.E.M. Significance of differences was calculated using Student’s *t* test (** *p* < 0.01; *p* ≥ 0.05).

**Figure 6 genes-13-00803-f006:**
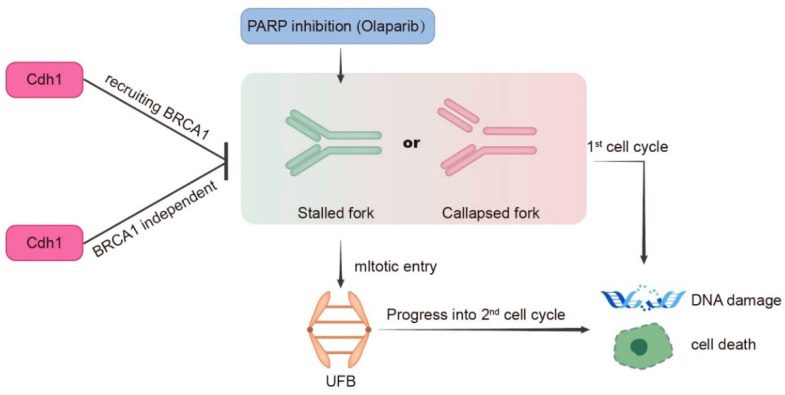
Model for Cdh1 function in genomic stability in response to PARP inhibition. During S phase, PARPi induces stalled or collapsed forks. Cdh1 prevents stalled or collapsed forks accumulation either through recruiting BRCA1 or its other functions that are not related to BRCA1. When Cdh1 is absent, excessive stalled or collapsed forks can directly lead to DNA damage and cell death in the first cell cycle; or upon mitotic entry, these abnormal replication structures generate ultrafine DNA bridges, which affects sister chromatids segregation leading to DNA damage and cell death in the next cell cycle.

**Table 1 genes-13-00803-t001:** shRNA sequences.

shRNA	Sequence
shNC	5′-TTCTCCGAACGTGTCACGT-3′
shCdh1-1	5′-TGAGAAGTCTCCCAGTCAG-3′
shCdh1-2	5′-GGATTAACGAGAATGAGAA-3′

## Data Availability

All data are contained within this article and available from the corresponding author on reasonable request.

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
