# Peer review of "Cdh1 Deficiency Sensitizes TNBC Cells to PARP Inhibitors"

_genes, 2022, doi:10.3390/genes13050803_

Round 1

Reviewer 1 Report

Li and colleagues in “Cdh1 deficiency sensitizes TNBC cells to PARP inhibitors” described the potential functions of Cdh1 in maintaining genomic stability in TNBC cells using bioinformatic and experimental technologies.

I have some concerns below:

  1. Method section should add the bioinformatic analysis details.
  2. In section 3.1, the authors should make sure if they’re using p value or adjusted p value for the cutoff, those two results may vary a lot, especially for the significant DE gene list, aslo the significant DE gene list should be added in the supplementary results.
  3. The informatic analysis results are inaccurate and make the basis of the manuscript less solid.

3.1 When authors extracted GSE data for the DESeq analysis, Fzr1 (Cdh1) gene was not the only or the top one of significantly differential expressed gene between tumor VS normal samples, and many other genes were also observed. However, the authors only focus Cdh1. 

3.2 It’s expected to see the how to correlate DNA damage repair with Cdh1 gene in breast cancer or triple negative cancer using the bioinformatic analysis, but the authors have not conducted any of such kind of analysis, and directly quote a sentence without any citation. Also from many previous studies, Cdh1 is not quietly closely related with DNA damage repair in TNBC.

3.3 “The bioinformatic analyses above and the observation that Cdh1 is involved in DNA damage repair prompted us to examine if its deficiency would sensitize breast cancer cells to PARP inhibitors.” Actually, the authors did not provide any results to correlate Cdh1 is involved in DNA damage repair, then the following analysis were not closely correlated with bioinformatic observations.

  1. As the introduction suggested, TNBC is very different from other breast cancer type, then why using publicly available breast cancer database but not only using TNBC to infer Cdh1 expression correlates with poor prognosis. In addition, the Discussion section motioned Cdh1 overexpression is correlated with poor prognosis in TNBC was not accurate.
  2. Did the authors calculate the p value for fig.2B and fig.2C.
  3. The author should add a schematic figure to descbribe the mechanism of Cdh1 function in genomic stability.
  4. The font format is inconsistence for the full manuscript. See details in Methods section
  5. Line142 with GSE datasets should add citations.
  6. Line150 replace “Having a p-value< 0.05” to “Having the adjusted p-value< 0.05”.

Author Response

Dear reviewer,

Thank you for your useful comments on our manuscript. We have modified the manuscript accordingly, and detailed corrections are listed below in a point-by-point manner. Please find the attachment.

Sincerely,

Wei Wu, PhD.

Zhejiang Provincial Key Laboratory of Pancreatic Disease, The First Affiliated Hospital of Zhejiang University,

Hangzhou, Zhejiang 310003, China.

Reviewer 2 Report

The experiments described in the manuscript „Cdh1 deficiency sensitizes TNBC cells to PARP inhibitors” seem well conducted; however, I still have some minor suggestions:

  1. English needs improving, which would help with the flow of the manuscript.
  2. An abstract is too descriptive and chaotic. There is no proper proportion between the sections: background, materials and methods, results, and conclusions.
  3. The last sentence in the Introduction section is unclear.
  4. The discussion section is very poor; it is not a real discussion with reviewing of other manuscripts on this topic, but a repetition of the results.
  5. There is no explanation for abbreviations first used throughout the manuscript.
  6. Try not to use abbreviations in the titles of the sections, e.g. 2.7. UFBs detection.
  7. Editorial errors occur, eg. P values (sometimes was written in uppercase sometimes in lowercase), s.e.m – should be written S.E.M. etc.

Round 2

Reviewer 1 Report

The authors have resolved my concerns except one point below:

3.2 It’s expected to see the how to correlate DNA damage repair with Cdh1 gene in breast cancer or triple negative cancer using the bioinformatic analysis, but the authors have not conducted any of such kind of analysis, and directly quote a sentence without any citation. Also from many previous studies, Cdh1 is not quietly closely related with DNA damage repair in TNBC.

When we tried to analyze differential genes in TNBC vs normal samples, we were pleasantly surprised to observe that FZR1 was on the list of 20-top up-regulated genes. Furthermore, KEGG pathway analysis indicated that the top irregulated genes in TNBC can positively regulate events associated with DNA replication and cell cycle checkpoints (Figure 2). Given that APC/Cdh1 is critical for DNA damage repair as we have shown previously, we hypothesize that TNBC cells lacking Cdh1 would be as sensitive to replication pressure as those cells without BRCA1.

--It’s not a good way to correlate DNA damage repair with Cdh1 gene. It would be better if the details of KEGG analysis noted here. KEGG is a gene clustering approach, and not directly associated with a specific gene. Albeit FZR1 was on the list of 20-top up-regulated genes, and KEGG clustering showing cell cycle checkpoint and DNA replication were top regulated pathway, and cannot infer that contributed by FZR1/Cdh1gene.
